# Prevalence of Vitamin B_12_ and Folate Deficiencies in Indian Children and Adolescents

**DOI:** 10.3390/nu15133026

**Published:** 2023-07-03

**Authors:** Tattari Shalini, Raghu Pullakhandam, Santu Ghosh, Bharati Kulkarni, Hemalatha Rajkumar, Harshpal S. Sachdev, Anura V. Kurpad, Geereddy Bhanuprakash Reddy

**Affiliations:** 1National Institute of Nutrition, Jamai-Osmania, Tarnaka, Hyderabad 500007, India; 2St. John’s Medical College, Bangalore 560034, India; santu.g@stjohns.in (S.G.);; 3Sitaram Bhartia Institute of Science and Research, New Delhi 110016, India

**Keywords:** vitamin B_12_ deficiency, folate deficiency, school-age children, adolescents, CNNS

## Abstract

Deficiencies of vitamin B_12_ (B_12_) and folate (FA) are of particular interest due to their pleiotropic role in 1-carbon metabolism. In addition to adverse birth outcomes, deficiencies of B_12_ and FA, or an imbalance in FA/B_12_ status, are linked to metabolic disorders. Indian diets that are predominantly plant food-based could be deficient in these vitamins, but there are no national estimates of the prevalence of B_12_ and FA deficiency in Indian children and adolescents, nor their associations with age, sex and growth indicators. The recent Comprehensive National Nutrition Survey (CNNS-2016-18) provided estimates of the prevalence of B_12_ and FA deficiency at the national and state levels among preschool (1–4 years: 9976 and 11,004 children, respectively), school-age children (5–9 years: 12,156 and 14,125) and adolescents (10–19 years: 11,748 and 13,621). Serum B_12_ and erythrocyte FA were measured by the direct chemiluminescence method and their deficiency was defined using WHO cut-offs. The prevalence of B_12_ and FA deficiency was high among adolescents (31.0%, CI: 28.7–33.5 and 35.6%, CI: 33.1–8.2) compared to school-age (17.3%, CI: 15.4–19.3 and 27.6%, CI: 25.5–29.9) and preschool children (13.8%, CI: 11.7–16.2 and 22.8%, CI: 20.5–25.2, respectively). The prevalence of both B_12_ and FA deficiency was significantly higher by 8% and 5%, respectively, in adolescent boys compared to girls. There was no association between anthropometric undernutrition and B_12_ and FA deficiency. There was wide regional variation in the prevalence of B_12_ and FA deficiency, but no rural–urban differences were observed across all age groups. The national prevalence of B_12_ deficiency among preschool or school-age children was <20% (the cut-off that indicates a public health problem). However, FA deficiency in these age groups and both FA and B_12_ deficiencies in adolescents were >20%, warranting further investigation.

## 1. Introduction

Among micronutrients, vitamin B_12_ (B_12_) and folate (FA) are critical as they are required in a plethora of metabolic and biological functions [1,2]. B_12_ and FA have overlapping biological functions in DNA synthesis and the development of red blood cells (RBC) and the myelin sheath, which are essential for normal growth and development [3]. One central pathway for both is the methyl transfer reaction in the methionine cycle, which converts homocysteine (Hcy) to methionine. Folate is engaged in many methylation reactions covering DNA, proteins, phospholipids and neurotransmitter metabolism [4]. B_12_ is only found in animal-source foods such as meat, poultry, fish and dairy products, while folate is abundant in both animal and plant foods. 

B_12_ and FA deficiencies occur throughout the lifecycle, with different outcomes. During pregnancy, they are associated with an increased risk of adverse outcomes such as neural tube defects and low birth weight, intrauterine growth retardation, miscarriage and pre-eclampsia [5,6,7]. In children, B_12_ and FA deficiency can result in megaloblastic anemia, poor growth and stunting, increased risk of infections, cognitive dysfunction, neurologic damage and brain atrophy in severe cases [2,3,4,8]. There is also a high prevalence of the double burden of malnutrition, where multiple biomarkers of cardiovascular disorders (CVDs) are elevated even in undernourished children and adolescents [9]. B_12_ and FA deficiencies are associated with hyperhomocysteinemia, which is a CVD risk factor [10,11,12], and during pregnancy, an imbalance in FA/B_12_ status has been associated with adverse birth outcomes [5] and adiposity and insulin resistance in the offspring [13]. Although a high prevalence of B_12_ (27–67%) and FA (12–42%) deficiency has been reported in India [11,12,14,15,16,17], the studies were not nationally representative.

The Indian Comprehensive National Nutrition Survey (CNNS) was conducted during 2016–2018 and evaluated the anthropometry, along with serum B_12_ and erythrocyte FA concentrations, among 1–19-year-old children and adolescents across all the Indian geographic states. This offered an opportunity to quantify the prevalence of B_12_ and FA deficiency at a national and state level in children and adolescents, stratified by age and gender. We also evaluated the association of B_12_ and FA deficiency with demographic and socioeconomic variables, as well as reported morbidity and anemia prevalence.

## 2. Methodology

### 2.1. CNNS Survey, Serum B_12_ and Erythrocyte FA Analysis

The CNNS was a community-based cross-sectional survey conducted among Indian children and adolescents in 29 states and the union territory of Delhi from February 2016 to October 2018 in collaboration with UNICEF, India and the Population Council, under the supervision of the Ministry of Health and Family Welfare, Government of India. The methodological details of the survey are available in the CNNS report [18]. Briefly, CNNS used a multi-stage stratified, probability proportional to size (PPS) cluster sampling to enroll preschool children (1–4 years), school-age children (5–9 years) and adolescents (10–19 years) to adequately represent the national, state, male–female and urban–rural population. For biological sampling, 50% of all the children who completed anthropometry were selected by systematic random sampling. Children/adolescents with a physical deformity, cognitive disability, chronic illness, acute febrile/infectious illness, acute injury or pregnancy were excluded. Ethical approval was obtained from the Institutional Review Boards of Population Council, New York, NY, USA, and the Post-Graduate Institute of Medical Education and Research, Chandigarh, India [18]. Informed consent from the parents/caregivers of children under 10 years, informed consent of parents/caregivers of adolescents (11–17 years) and the latter’s agreement, and informed consent of adolescents older than 17 years were obtained. All procedures and methods were performed in accordance with the Declaration of Helsinki.

Household socioeconomic and demographic characteristics, information on history of morbidity in the preceding two weeks and iron–folic acid (IFA) supplementation in the previous week, and anthropometric data of one child/adolescent per age group were collected from each household. The wealth index based on the possession of common household items and facilities was computed as described in the National Family Health Survey (NFHS)-4 [19]. Access to facilities such as drinking water, hand washing and sanitation was categorized based on WHO/UNICEF Joint Monitoring Programme for Water Supply, Sanitation and Hygiene (WASH) guidelines [20]. Age–sex standardized height-for-age (HAZ), weight-for-height (WHZ), and BMI-for-age Z-scores were calculated using the WHO Growth Reference Standards [21].

The day before sample collection, parents and children were instructed to ensure overnight fasting (8–10 h) in the latter. Venous blood samples (7 mL blood from children under 5 and 10 mL from 5–19-year-olds) and recordings of binary (yes/no) information on fasting status and time of sample collection were obtained by trained phlebotomists. The blood samples were transported in cool bags (3L-12H-08P, PronGo) to the nearest collection center, where the serum/plasma and erythrocytes were separated and divided into aliquots within 6 h of sample collection. Biochemical analyses were carried out by SRL Labs in Mumbai, Gurugram and Kolkata, India, and are reported in detail elsewhere [22]. Briefly, serum B_12_ and erythrocyte FA levels were estimated using a chemiluminescence-based competitive immunoassay (Siemens Centaur) [18]. Hemoglobin was estimated in whole blood by the cyanmethemoglobin method (Beckman Coulter, LH 750). Rigorous quality control procedures were implemented for sample collection, transportation and testing using standard internal and external quality assurance procedures [18,22].

Using WHO guidelines, B_12_ deficiency was defined as serum B_12_ < 203 pg/mL (150 pmol/L; conversion factor: 0.738) and FA deficiency as erythrocyte FA < 151 ng/mL (342 nmol/L; conversion factor: 2.266) for all age groups [23]. Anemia was diagnosed using WHO Hb cut-offs (g/dL): <11.0 (1–4 years), <11.5 (5–11 years), <12.0 (12–14 years), <12.0 (15–19 years, girls) and <13.0 (15–19 years, boys) [24]. According to the Biomarkers Reflecting Inflammation and Nutritional Determinants of Anemia (BRINDA) report, B_12_ and FA concentrations do not require adjustment for inflammation [25], and hence, no adjustment was performed in the present study. 

### 2.2. Statistical Analyses

Statistical analyses were conducted using the SPSS statistical package (version 23, SPSS Inc., Chicago, IL, USA). The proportion of demographic characteristics of the study sample included in the present analysis was compared with the proportion in the entire CNNS survey sample to rule out selection bias due to nested sampling. Serum B_12_ and erythrocyte FA concentrations are presented as geometric mean (GM) and geometric standard deviation (GSD), since their distributions were skewed. Relevant sampling weights were used wherever indicated in order to ensure the representativeness of the estimates at the national/state level as well as at the local level, such as rural, urban and urban slum areas in metropolitan cities. The prevalence of B_12_ and FA deficiency, along with 95% confidence intervals (CI), was estimated at the national and state levels. Sub-group analyses were also performed to evaluate urban–rural, age, gender, sociodemographic and WASH differentials. The association between the prevalence of B_12_ and FA deficiency with age groups in different states was evaluated using the Spearman rank-order correlation.

## 3. Results

### 3.1. Characteristics of the Study Population

A total of 1,05,243 children and adolescents (preschool: 31,058, school-age: 38,355, adolescents: 35,830) were interviewed and their anthropometric data were collected; serum B_12_ and erythrocyte FA concentrations were available for 33,880 and 38,750 children and adolescents (preschool: 9976 and 11,004, school-age: 12,156 and 14,125 and adolescents: 11,748 and 13,621, respectively) (Figure 1). The sociodemographic characteristics were almost similar among participants for whom anthropometric data were collected (total sample) and the study sample (B_12_ and FA), except the proportion of children aged 3–4 and 7–9 years included in the study sample was higher than that of those aged 1–2 and 5–6 years (61% vs. 39%) (Appendix A). Table 1 shows the age-specific general characteristics of the study population. Among preschool children, 35% were stunted and underweight, 16% were wasted/thin and about 15% had diarrhea two weeks prior to the survey in both B_12_ and FA study samples.

### 3.2. Serum B_12_ and Erythrocyte FA Concentration and Prevalence of B_12_ and FA Deficiency by Age and Sex

The GMs of serum B_12_ (pg/mL) and erythrocyte FA (ng/mL) concentration were significantly different among preschoolers, school-age children and adolescents (Table 2A,B). The national prevalence of B_12_ and FA deficiency was higher among adolescents (31.0%, CI: 28.7–33.5 and 35.6%, CI: 33.1–8.2) compared to school-age (17.3%, CI: 15.4–19.3 and 27.6%, CI: 25.5–29.9) and preschool children (13.8%, CI: 11.7–16.2 and 22.8%, CI: 20.5–25.2, respectively) (Table 2A,B). Though B_12_ and FA concentrations tended to decline with age (1–19 years) in both genders, the decline was significantly greater in adolescent boys compared to girls (Table 2A,B). As a consequence, the prevalence of B_12_ and FA deficiency increased with age (Figure 2), and adolescent boys had 8% higher B_12_ and 5% higher FA deficiency compared to girls (Table 2A,B).

### 3.3. State-Based, Rural–Urban and Regional Differences in Prevalence of B_12_ and FA Deficiency

The point prevalence of B_12_ deficiency varied across states, being the highest in Gujarat (preschoolers—29.2%, CI: 20.3–40.0; adolescents—47.6%, CI: 37.3–58.2) and Punjab (school-age children—32.4%, CI: 25.2–40.5) and the lowest in West Bengal (preschoolers—1.9%, CI: 0.4–8.5) and Kerala (school-age children—0.9%, CI: 0.2–3.6; adolescents—2.3%, CI: 1.0–5.5) (Figure 3). Similarly, the point prevalence of FA deficiency was the highest in Nagaland (preschoolers—71.1%, CI: 55.5–83.0; adolescents—85.9%, CI: 63.3–94.9) and Andhra Pradesh (school-age children—67.8%, CI: 60.4–74.3) and the lowest in Sikkim for all age groups (0.1–0.8%) (Figure 3).

A significant positive relationship of B_12_ and FA deficiency prevalence between the age groups by state was noted (1–4 vs. 5–9 years: r = 0.888, *p* < 0.001 and r = 0.961, *p* < 0.001; 1–4 vs. 10–19 years: r = 0.747, *p* < 0.001 and r = 0.936, *p* < 0.001; 5–9 vs. 10–19 years: r = 0.938, *p* < 0.001 and r = 0.967, *p* < 0.001, respectively) (Appendix A).

Further, there was wide regional variation in the prevalence of B_12_ and FA deficiency. While the prevalence of B_12_ deficiency was high in the central region across all the age groups (preschoolers—21.0%, CI: 14.9–28.7; school-age children—29.1%, CI: 24.5–34.2; adolescents—42.7%, CI: 37.4–48.2), the prevalence of FA deficiency was high in the northeast for preschoolers (47.4%, CI: 39.1–55.8), the west for school-age children (54.9%, CI: 50.2–59.5) and the south for adolescents (71.6%, CI: 66.8–76.0) (Figure 4). However, the B_12_ and FA prevalence was similar between rural and urban locations across all age groups (Appendix A).

### 3.4. B_12_ and FA Deficiency by Sociodemographics, WASH Characteristics, Undernutrition and Morbidity

The prevalence of B_12_ deficiency was higher in school-age children of mothers who had lower education (Appendix A). While no association was observed between B_12_ deficiency and WASH variables, the prevalence of FA deficiency was higher with unimproved drinking water among all the age groups (preschoolers: 35.8%, CI: 23.0–51.0, school-age children: 39.2%, CI: 30.4–48.8, adolescents: 54.7%, CI: 45.7–63.5). No association was observed between B_12_ deficiency and the wealth index across age groups. However, children and adolescents (5–19 years) from richer households (school-age children: 29.6%, CI: 26.6–32.8 and adolescents: 42.0%, CI: 37.2–47.0) had higher FA deficiency than those from poorer households (school-age children: 20.7%, CI: 15.4–27.1 and adolescents: 27.0%, CI: 21.6–33.2) (Appendix A).

There was no association between anthropometric undernutrition and B_12_ and FA deficiency, except for a significantly lower prevalence of B12 deficiency among 5–9-year-old severely stunted children and 10–19-year-old severely wasted children (Table 3). Children with diarrhea (preschoolers: 14.9%, CI: 11.5–19.1 and school-age children: 20.2%, CI: 15.8–25.5) and fever (school-age children: 22.8%, CI: 19.6–26.4) in the two weeks preceding the survey had significantly lower FA deficiency than those without morbidity, while no association was found between B_12_ deficiency and morbidity (Table 3). Both serum B_12_ and erythrocyte FA concentrations were negatively associated with hemoglobin levels. In preschool-age children, FA deficiency was higher in those who did not receive the IFA supplement compared to those who had received it in the previous week (Table 4).

## 4. Discussion

This study provides the serum B_12_ and erythrocyte FA levels and their prevalence estimates in a representative sample of Indian children and adolescents at the national, state and regional levels. The prevalence of B_12_ deficiency was high among adolescents (31%), with ~50% lower prevalence in preschool (13.8%) and school-age (17.3%) children. Similarly, the prevalence of FA deficiency was also higher in adolescents (35.6%) compared to preschool (22.8%) and school-age (27.6%) children. 

In the present study, the prevalence of B_12_ deficiency in children and adolescents was lower, while the prevalence of FA deficiency was almost similar to estimates from previous Indian and other studies [14,15,26,27,28]. However, except for the very recent study by Awasthi et al. [29], these previous studies had small sample sizes and were not nationally representative. Interestingly, a recent nationally representative study conducted soon after this CNNS study (2016–18) also reported almost similar trends of B_12_ deficiency, but FA deficiency was slightly lower. Further, similar to the findings reported in the present study, Awasthi et al. [29] also found a higher prevalence of B_12_ and FA deficiency with increased age, and more so in boys. However, this study is not state-representative and did not capture the regional differences and associated factors. While some nationally representative surveys in other countries have also demonstrated a low prevalence of B_12_ deficiency among 1–6-year-old children in Mexico (7.7%) [30] and school-aged children in Venezuela (11.4%) [31,32], a Guatemalan study reported the prevalence of B_12_ and FA deficiency to be 22.5% and 33.5%, respectively, among children aged 6–59 months [7]. In countries where animal foods constitute ~5–10% of the energy intake, the prevalence of B_12_ deficiency was high. For example, the prevalence of B_12_ deficiency was >70% in school children in Kenya [33] and 27% in preschoolers in New Delhi [14]. A study in Nepal revealed 41% B_12_ deficiency (serum B_12_ < 150 pmol/L) plus 16% depletion (150–200 pmol/L) in 6–35-month-old children with acute diarrhea [26]. Our recent studies among apparently healthy adults showed a high prevalence of B_12_ deficiency (~40%) that is associated with suboptimal dietary intakes [11,16]. 

The differences in the magnitude of the prevalence of deficiency among the reported studies, including the present study, might be multifactorial, including methodological issues. For example, in the case of FA, the data availability is complicated due to the large differences in the assay approaches (e.g., microbiological, immunoassay, and chromatography-based), analytes (e.g., total compared with major circulating types of folate) and antibodies used for immunoassay approaches, and because of the measurement of FA in serum or RBC. In addition, biomarkers such as serum methylmalonic acid (MMA) and Hcy could be more sensitive measures of B_12_ deficiency [34]. Serum holotranscobalamin concentration is an early marker with a better representation of the actual B_12_ status [35,36]. In a recent study in apparently healthy adults, we found a higher prevalence of B_12_ deficiency (46%) with holotranscobalamin compared to total B_12_ measurement (37%) [12].

Although deficiency of these vitamins can occur primarily as a result of insufficient dietary intake or malabsorption, various other factors such as gender, age and genetic, ethnic and sociocultural backgrounds are likely to influence their status [37]. Further, diets that are predominantly cereal-based and low in vegetables and fruits or animal-source food could contribute to the deficiency of these vitamins. The prevalence of B_12_ deficiency in the present study was higher in school-age children of mothers who had lower education. Similarly, the FA prevalence was higher among the participants with unimproved drinking water. However, intriguing trends were observed with a higher prevalence of FA deficiency in participants from higher socioeconomic status (SES) households (indicated by richer wealth quintiles than those from the lower SES. Nevertheless, a similar pattern of lower B_12_ deficiency in the low SES group was reported by a study on rural school children in Raigad, India [38]. 

Rural or urban residence represents an aggregate of multiple factors, with rural residents more likely to be from low SES households with poorer WASH facilities. Although the overall prevalence of B_12_ and FA deficiency was similar between urban and rural participants, significant regional differences were observed. The central region showed higher B_12_ prevalence across all age groups. Likewise, the state-wise prevalence of B_12_ and FA deficiency showed perplexing trends, with relatively richer states (in terms of per capita income as well consumption of milk and dairy products) such as Punjab and Gujarat showing higher prevalence. The inverse socioeconomic gradients of B_12_ and FA prevalence are difficult to explain and need further exploration.

Although the lower prevalence of B12 deficiency in severely stunted (5–9 years) and wasted (10–19 years) children is counter-intuitive, this could be a ramification of reduced metabolic requirements in the face of reduced growth and/or muscular activity. Further, in the frame of anemia, B_12_ and FA deficiency were earlier shown to be associated with 19–25% of anemia prevalence in children and adolescents in this survey [39]. In a more recent study on the same datasets, it was found that FA deficiency was negatively associated with anemia while vitamin B12 deficiency was not associated with anemia [40]. In addition to increased growth requirements in children and adolescents, chronic low intakes through predominantly vegetarian diets and poor absorption could induce the risk of these vitamin deficiencies [41].

Adolescent boys had a higher prevalence of B_12_ deficiency compared to girls, which is in line with a previous study in Venezuela, where the B_12_ deficiency prevalence was higher in adolescents compared to infants and children [31]. Similarly, other studies in adult population showed a higher prevalence of vitamin B_12_ deficiency in males compared to females, commensurate with higher Hcy in males [11,12,16]. A higher prevalence of vitamin B_12_ deficiency among boys may be explained by a higher requirement for micronutrients to sustain rapid muscular growth during adolescence as compared to girls. Furthermore, in a separate regional study, dietary vitamin B_12_ intake was lower among boys compared to girls [17]. The inverse relation between male sex or age with vitamin B_12_ status could be interpreted in light of greater requirements for more rapid growth in boys than girls and in older than younger children that are not being met with adequate dietary intake. At least in Colombia, older children were shown to be less likely to adhere to an animal protein intake pattern, which supports this possibility [42], whereas an Indian study found that rural youth with the lowest plasma vitamin B12 levels infrequently or never consumed dairy products/non-vegetarian foods [43]. 

The strengths of our study include a large sample, representative at regional, state and national levels, covering a wide range of age (1–19 years), and the estimates provided of the prevalence of B_12_ and FA deficiency. The important limitations include smaller sample sizes in some of the states and a lower proportion of 1–2-year-old children in the study sample, which may have resulted in the underestimation of B_12_ and FA deficiency prevalence in the 1–4-year-old age group. Another limitation pertains to the lack of data on other more sensitive biomarkers of B_12_ deficiency such as MMA, Hcy and holotranscobalamin.

## 5. Conclusions

Our study bridges a critical information gap on the prevalence of B_12_ and FA deficiency in Indian children and adolescents and demonstrates that about a third of adolescent boys are likely to be deficient in B_12_ and FA. The prevalence of these deficiencies, however, is lower in younger age groups. These findings are important to inform nutrition policy in India. More consistent use of thresholds to define deficiency is needed in order to assess the realistic public health significance of FA and B_12_ deficiency.

## Figures and Tables

**Figure 1 nutrients-15-03026-f001:**
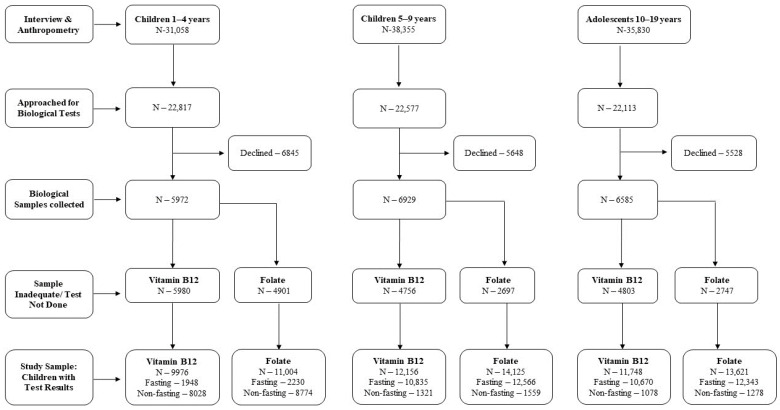
Flow chart for the recruitment of participants in the CNNS survey and the selection of samples for vitamin B_12_ and folate analyses.

**Figure 2 nutrients-15-03026-f002:**
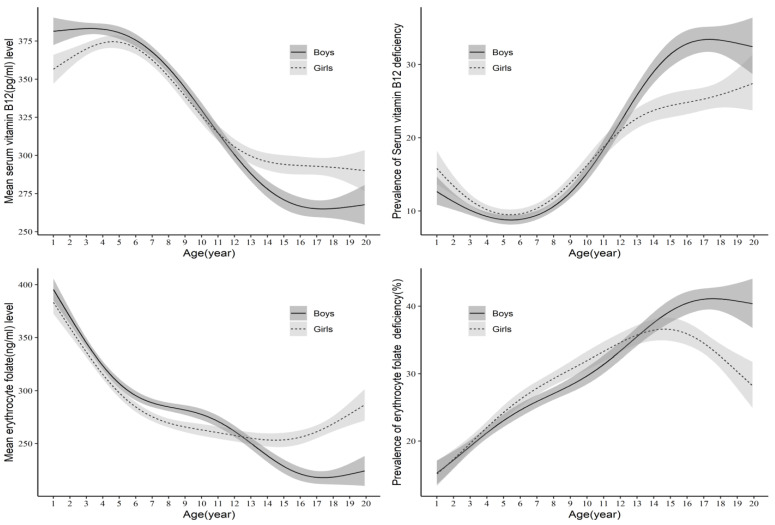
Serum vitamin B_12_ and erythrocyte folate concentrations (**left**) and prevalence of vitamin B_12_ and folate deficiency (**right**) as a function of age and gender. The line indicates the geometric mean and the shaded areas are 95% confidence bands.

**Figure 3 nutrients-15-03026-f003:**
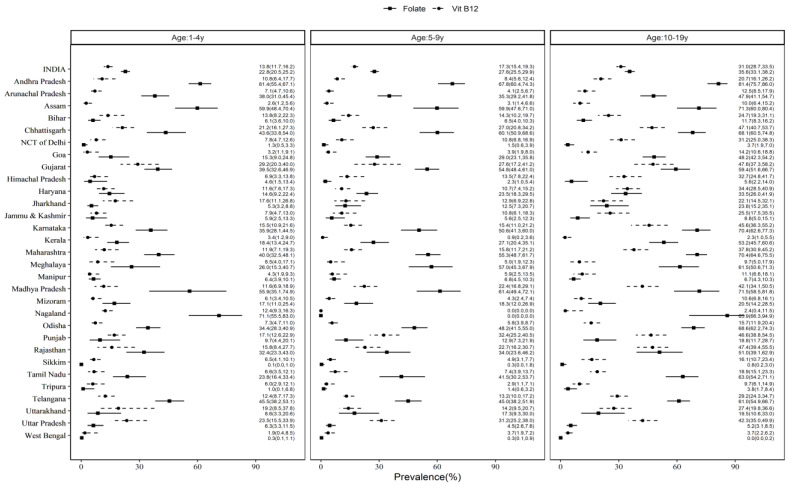
Prevalence of vitamin B_12_ and folate deficiency in children and adolescents across geographical states of India. The dots indicate means and bars are 95% CI. NCT: National Capital Territory.

**Figure 4 nutrients-15-03026-f004:**
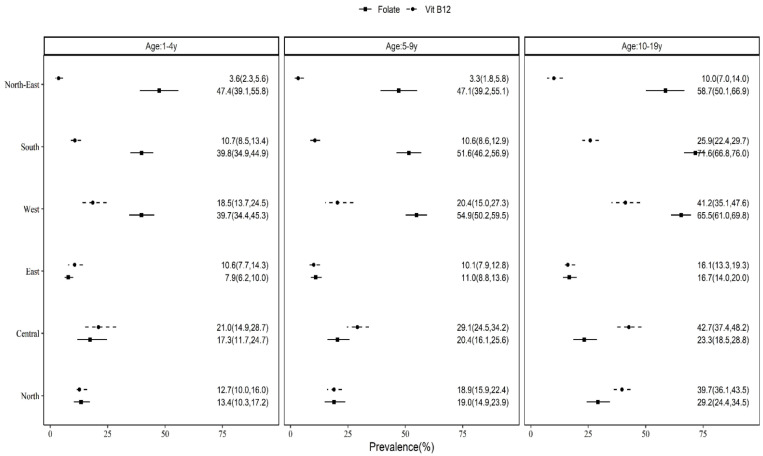
Prevalence of vitamin B_12_ and folate deficiency by region in children and adolescents. The dots indicate means and bars are 95% CI.

**Table 1 nutrients-15-03026-t001:** Characteristics of the study population.

Characteristics		1–4 Years		5–9 Years		10–19 Years	
		Vitamin B12 (*n* = 9976) % (95% CI)	Folate (*n* = 11,004) % (95% CI)	Vitamin B12 (*n* = 12,156) % (95% CI)	Folate (*n* = 14,125) % (95% CI)	Vitamin B12 (*n* = 11,748) % (95% CI)	Folate (*n* = 13,621) % (95% CI)
Age in years	Mean (95% CI)	2.78 (2.74–2.83)	2.78 (2.74–2.83)	7.02 (6.97–7.06)	7.02 (6.98–7.07)	14.3 (14.2–14.4)	14.3 (14.2–14.4)
Sex	Boys	52.9 (50.3–55.4)	52.5 (50.0–55.0)	51.6 (49.8–53.3)	51.2 (49.4–52.9)	50.6 (48.6–52.5)	50.6 (48.8–52.4)
	Girls	47.1 (44.6–49.7)	47.5 (45.0–50.0)	48.4 (46.7–50.2)	48.8 (47.1–50.6)	49.4 (47.5–51.4)	49.4 (47.6–51.2)
Residence	Urban	24.9 (21.7–28.4)	25.7 (22.5–29.1)	23.6 (20.7–26.8)	25.1 (22.1–28.2)	25.2 (22.1–28.6)	25.7 (22.6–29.0)
	Rural	75.1 (71.6–78.3)	74.3 (70.9–77.5)	76.4 (73.2–79.3)	74.9 (71.8–77.9)	74.8 (71.4–77.9)	74.3 (71.0–77.4)
Wealth Index	Poorest	15.9 (13.8–18.3)	15.8 (13.7–18.1)	18.1 (16.2–20.3)	18.0 (15.8–20.4)	17.8 (15.5–20.4)	18.3 (16.0–20.8)
	Poor	21.2 (18.5–24.1)	20.5 (17.9–23.4)	21.3 (19.4–23.3)	20.2 (18.5–21.9)	20.5 (18.8–22.3)	20.6 (18.9–22.3)
	Middle	22.3 (20.4–24.3)	21.7 (19.9–23.5)	21.5 (19.9–23.3)	21.0 (19.4–22.5)	21.4 (19.8–23.1)	20.3 (18.8–21.9)
	Rich	20.8 (18.9–22.9)	21.7 (19.7–23.9)	21.0 (19.4–22.7)	21.6 (19.9–23.3)	20.9 (19.3–22.7)	21.0 (19.4–22.8)
	Richest	19.8 (17.7–22.0)	20.3 (18.3–22.5)	18.0 (16.4–19.8)	19.3 (17.6–21.1)	19.3 (17.5–21.2)	19.8 (17.9–21.8)
Mother’s Schooling	Primary	34.3 (31.7–36.9)	34.6 (31.9–37.3)	47.8 (45.4–50.3)	47.9 (45.6–50.3)	16.3 (14.3–18.4)	15.9 (14.1–18.0)
	Secondary	44.3 (41.8–46.8)	43.6 (41.3–46.0)	40.1 (38.0–42.3)	39.8 (37.8–41.8)	68.7 (66.2–71.1)	69.3 (67.0–71.5)
	Higher secondary	10.8 (9.3–12.5)	11.4 (9.9–13.1)	6.7 (6.0–7.6)	6.9 (6.2–7.8)	9.7 (7.7–12.1)	9.5 (7.7–11.6)
	Graduation and above	10.7 (9.2–12.4)	10.4 (9.0–12.0)	5.3 (4.7–6.1)	5.4 (4.8–6.1)	5.3 (4.4–6.3)	5.3 (4.5–6.3)
Father’s Occupation	Professional	7.9 (6.8–9.2)	8.5 (7.4–9.9)	9.4 (8.1–10.8)	9.5 (8.3–10.8)	10.0 (8.5–11.7)	9.4 (8.0–10.9)
	Sales and services	26.8 (24.6–29.1)	28.0 (25.7–30.4)	23.1 (21.4–25.0)	24.2 (22.3–26.1)	24.4 (22.7–26.2)	24.1 (22.5–25.8)
	Manual, agriculture	51.5 (48.8–54.1)	50.4 (47.6–53.2)	54.8 (52.2–57.4)	53.8 (51.4–56.2)	51.3 (48.8–53.7)	52.4 (50.0–54.7)
	Others	13.8 (11.8–16.0)	13.1 (11.3–15.1)	12.7 (11.1–14.5)	12.5 (11.0–14.2)	14.3 (12.5–16.4)	14.2 (12.4–16.1)
Child schooling	Yes	-	-	92.2 (91.1–93.2)	92.1 (91.1–93.0)	80.7 (78.9–82.4)	80.9 (79.2–82.5)
	No	-	-	7.8 (6.8–8.9)	7.9 (7.0–8.9)	19.3 (17.6–21.1)	19.1 (17.5–20.8)
Stunting	No stunting (HAZ < −2SD)	64.4 (62.1–66.7)	64.5 (62.3–66.7)	79.3 (77.7–80.8)	78.6 (76.9–80.3)	73.0 (71.0–74.9)	73.4 (71.6–75.2)
	Moderate (HAZ: −3 to −2SD)	24.0 (22.2–26.0)	23.3 (21.6–25.0)	15.5 (14.3–16.8)	16.4 (15.1–17.8)	21.3 (19.6–23.1)	20.7 (19.2–22.3)
	Severe (HAZ < −3SD)	11.5 (10.1–13.2)	12.2 (10.7–13.9)	5.1 (4.3–6.1)	5.0 (4.2–5.8)	5.7 (4.9–6.6)	5.9 (5.1–6.7)
Underweight	Not present (WAZ < −2SD)	64.8 (62.0–67.6)	65.1 (62.5–67.6)	-	-	-	-
	Moderate (WAZ: −3 to −2SD)	26.2 (23.7–28.8)	26.7 (24.3–29.2)	-	-	-	-
	Severe (WAZ < −3SD)	9.0 (7.8–10.3)	8.2 (7.2–9.4)	-	-	-	-
Wasting/ Thinness	Not present (WHZ < −2SD)	84.0 (82.1–85.6)	85.0 (83.5–86.5)	76.4 (74.8–78.0)	76.6 (75.2–78.0)	75.5 (73.7–77.2)	75.7 (74.0–77.2)
	Moderate (WHZ:−3 to −2SD)	12.2 (10.8–13.8)	11.6 (10.4–12.9)	18.5 (17.1–19.9)	18.4 (17.1–19.8)	18.1 (16.6–19.7)	17.9 (16.6–19.4)
	Severe (WHZ < −3SD)	3.8 (3.0–4.8)	3.4 (2.8–4.1)	5.1 (4.3–6.0)	5.0 (4.3–5.8)	6.4 (5.7–7.3)	6.4 (5.6–7.3)
Drinking water source	Piped and improved	85.0 (82.4–87.3)	85.2 (82.6–87.5)	84.9 (82.5–87.0)	85.2 (82.9–87.2)	85.6 (83.5–87.5)	86.2 (84.2–87.9)
	Non-piped and improved	8.9 (7.0–11.2)	9.2 (7.4–11.3)	8.2 (6.6–10.2)	8.1 (6.7–9.9)	8.1 (6.8–9.6)	7.7 (6.5–9.1)
	Unimproved	6.1 (4.8–7.6)	5.6 (4.3–7.4)	7.0 (5.7–8.4)	6.7 (5.3–8.4)	6.3 (5.1–7.9)	6.1 (5.0–7.6)
Handwashing	Basic	50.3 (47.5–53.1)	52.4 (49.7–55.2)	46.8 (44.1–49.5)	49.3 (46.7–51.9)	47.8 (45.3–50.3)	48.7 (46.2–51.3)
	Limited	36.1 (33.4–38.8)	33.8 (31.3–36.5)	39.4 (36.6–42.3)	37.2 (34.6–39.8)	35.5 (33.1–38.0)	34.7 (32.5–37.1)
	No facility	13.6 (11.7–15.9)	13.7 (11.8–15.9)	13.8 (12.1–15.8)	13.5 (11.8–15.4)	16.7 (14.7–19.0)	16.5 (14.4–18.8)
Sanitation	Improved and not shared	43.8 (40.5–47.2)	44.1 (41.0–47.1)	39.7 (37.2–42.3)	40.5 (38.0–43.0)	47.8 (45.3–50.4)	47.1 (44.5–49.7)
	Improved and shared	12.2 (10.8–13.7)	13.0 (11.6–14.6)	12.3 (11.0–13.8)	12.3 (11.0–13.8)	8.7 (7.7–9.8)	9.0 (8.0–10.1)
	Unimproved	44.0 (40.1–48.0)	43.0 (39.3–46.7)	48.0 (44.8–51.2)	47.2 (44.1–50.4)	43.5 (40.6–46.4)	43.9 (41.1–46.8)
History of diarrhea in the two weeks prior to survey	Yes	15.0 (12.9–17.5)	15.3 (13.3–17.6)	9.4 (8.2–10.7)	9.0 (8.0–10.2)	-	-
	No	85.0 (82.5–87.1)	84.7 (82.4–86.7)	90.6 (89.3–91.8)	91.0 (89.8–92.0)	-	-
History of fever in the two weeks prior to survey	Yes	30.7 (28.3–33.2)	31.5 (29.2–33.9)	21.8 (19.8–24.0)	22.2 (20.5–24.0)	-	-
	No	69.3 (66.8–71.7)	68.5 (66.1–70.8)	78.2 (76.0–80.2)	77.8 (76.0–79.5)	-	-

HAZ; height-for-age, WHZ; weight-for-height, WAZ; weight-for-age.

**Table 2 nutrients-15-03026-t002:** (**A**) Serum vitamin B_12_ levels and prevalence of vitamin B_12_ deficiency (**B**) Erythrocyte folate levels and prevalence of folate deficiency among children and adolescents stratified based on sex and age groups.

**(A)**						
**Sex**	**1–4 Years (*n* = 9976)**		**5–9 Years (*n* = 12,156)**		**10–19 Years (*n* = 11,748)**	
	**Vitamin** **B_12_** **(pg/mL)** **Geometric Mean (95% CI)**	**Vitamin** **B_12_** **Deficiency % (95% CI)**	**Vitamin** **B_12_** **(pg/mL)** **Geometric Mean (95% CI)**	**Vitamin** **B_12_** **Deficiency % (95% CI)**	**Vitamin** **B_12_** **(pg/mL)** **Geometric Mean (95% CI)**	**Vitamin** **B_12_** **Deficiency % (95% CI)**
Boys	310.3 ^a^ (301.4–319.3)	14.3 ^a^ (11.4–17.7)	297.3 ^a^ (290.0–304.7)	16.7 ^a^ (14.7–18.9)	241.6 ^a^ (235.2–248.2)	35.0 ^a^ (31.8–38.3)
Girls	313.3 ^a^ (304.1–322.8)	13.3 ^a^ (10.9–16.2)	291.7 ^a^ (283.5–300.2)	17.9 ^a^ (15.3–20.8)	257.1 ^b^ (250.6–263.8)	27.0 ^b^ (24.4–29.7)
Total	311.7 * (305.0–318.5)	13.8 ^†^ (11.7–16.2)	294.6 ^#^ (288.2–301.0)	17.3 ^†^ (15.4–19.3)	249.2 ^$^ (243.9–254.6)	31.0 ^‡^ (28.7–33.5)
**(B)**						
**Sex**	**1–4 Years (*n* = 11,004)**		**5–9 Years (*n* = 14,125)**		**10–19 Years (*n* = 13,621)**	
	**Folate (ng/mL)** **Geometric Mean** **(95% CI)**	**Folate** **Deficiency %** **(95% CI)**	**Folate (ng/mL)** **Geometric Mean (95% CI)**	**Folate** **Deficiency % (95% CI)**	**Folate (ng/mL)** **Geometric Mean (95% CI)**	**Folate** **Deficiency %** **(95% CI)**
Boys	245.8 ^a^ (233.4–258.9)	22.6 ^a^ (19.7–25.8)	217.7 ^a^ (208.0–228.0)	27.7 ^a^ (25.1–30.4)	173.3 ^a^ (164.4–182.6)	38.3 ^a^ (35.3–41.4)
Girls	237.8 ^a^ (225.1–251.1)	22.9 ^a^ (20.2–25.9)	212.2 ^a^ (201.8–223.2)	27.6 ^a^ (25.1–30.3)	196.7 ^b^ (186.6–207.3)	32.9 ^a^ (30.2–35.7)
Total	241.9 * (231.4–252.9)	22.8 ^†^ (20.5–25.2)	215.0 ^#^ (206.4–224.1)	27.6 ^‡^ (25.5–29.9)	184.5 ^$^ (176.3–193.0)	35.6 ^§^ (33.1–38.2)

Superscripts ^ab^ in the same column indicate estimates with non-overlapping CIs. Superscripts ***^#^**^$^ and **^†‡^** in the same row for serum vitamin B12 levels and vitamin B12 deficiency, and superscripts ***^#^**^$^ and **^†‡§^** in the same row for erythrocyte folate levels and folate deficiency, respectively.

**Table 3 nutrients-15-03026-t003:** Prevalence of vitamin B12 and folate deficiency by under-nutrition and morbidity variable.

Characteristics		1–4 Years		5–9 Years		10–19 Years	
		Vitamin B12 Deficiency % (95% CI)	Folate Deficiency % (95% CI)	Vitamin B12 Deficiency % (95% CI)	Folate Deficiency % (95% CI)	Vitamin B12 Deficiency % (95% CI)	Folate Deficiency % (95% CI)
Stunting	No Stunting (HAZ < −2 SD)	12.7 ^a^ (10.2–15.8)	22.9 ^a^ (20.5–25.5)	17.8 ^a^ (15.8–20.0)	28.1 ^a^ (25.9–30.4)	31.2 ^a^ (28.8–33.7)	37.2 ^a^ (34.4–40.1)
	Moderate (HAZ: −3 to −2 SD)	15.1 ^a^ (12.2–18.5)	23.5 ^a^ (19.9–27.5)	18.9 ^a^ (15.2–23.2)	27.2 ^a^ (23.4–31.4)	31.5 ^a^ (26.3–37.3)	31.8 ^a^ (27.9–35.9)
	Severe (HAZ < −3 SD)	18.9 ^a^ (12.7–27.3)	21.6 ^a^ (16.8–27.4)	8.2 ^b^ (5.2–12.8)	22.2 ^a^ (16.7–28.8)	27.6 ^a^ (20.3–36.3)	29.1 ^a^ (23.5–35.5)
Wasting/Thinness	Not present (WAZ < −2 SD)	14.8 ^a^ (12.3–17.6)	22.8 ^a^ (20.3–25.5)	18.0 ^a^ (15.9–20.3)	27.4 ^a^ (25.1–29.9)	32.9 ^a^ (30.0–35.9)	35.8 ^a^ (32.9–38.7)
	Moderate (WAZ: −3 to −2 SD)	11.1 ^a^ (8.2–14.8)	21.5 ^a^ (17.5–26.2)	17.0 ^a^ (13.7–20.8)	29.1 ^a^ (25.6–32.9)	26.8 ^ac^ (23.0–30.8)	35.0 ^a^ (31.1–39.1)
	Severe (WAZ < −3 SD)	7.8 ^a^ (4.7–12.7)	24.0 ^a^ (17.1–32.7)	12.1 ^a^ (8.0–18.0)	26.4 ^a^ (20.5–33.4)	21.6 ^bc^ (16.6–27.5)	34.0 ^a^ (28.4–40.1)
Underweight	Not present (WHZ < −2 SD)	13.6 ^a^ (11.1–16.5)	23.1 ^a^ (20.5–25.9)	-	-	-	-
	Moderate (WHZ: −3 to −2 SD)	15.8 ^a^ (12.2–20.2)	22.6 ^a^ (18.9–26.8)	-	-	-	-
	Severe (WHZ < −3 SD)	11.8 ^a^ (8.1–16.9)	20.4 ^a^ (16.5–24.9)	-	-	-	-
History of diarrhea in the two weeks prior to survey	Yes	19.1 ^a^ (12.1–28.8)	14.9 ^a^ (11.5–19.1)	23.8 ^a^ (18.1–30.5)	20.2 ^a^ (15.8–25.5)	-	-
	No	12.9 ^a^ (11.1–15.0)	24.2 ^b^ (21.8–26.7)	16.6 ^a^ (14.8–18.6)	28.4 ^b^ (26.1–30.7)	-	-
History of fever in the two weeks prior to survey	Yes	16.9 ^a^ (11.6–23.9)	19.6 ^a^ (16.4–23.3)	15.8 ^a^ (12.5–19.8)	22.8 ^a^ (19.6–26.4)	-	-
	No	12.5 ^a^ (10.7–14.4)	24.2 ^a^ (21.5–27.1)	17.7 ^a^ (15.9–19.7)	29.0 ^b^ (26.7–31.4)	-	-

Superscripts ^abc^ in the same column indicate estimates with non-overlapping CIs. HAZ; height-for-age, WHZ; weight-for-height, WAZ; weight-for-age.

**Table 4 nutrients-15-03026-t004:** Prevalence of vitamin B12 and folate deficiency with iron–folic acid supplementation in all age groups.

IFA	1–4 Years		5–9 Years		10–19 Years	
	Vitamin B12 Deficiency % (95% CI)	Folate Deficiency % (95% CI)	Vitamin B12 Deficiency % (95% CI)	Folate Deficiency % (95% CI)	Vitamin B12 Deficiency % (95% CI)	Folate Deficiency % (95% CI)
Yes	11.2 ^a^ (7.2–16.9)	14.6 ^a^ (9.9–21.0)	18.1 ^a^ (13.8–23.2)	28.9 ^a^ (23.4–35.1)	28.5 ^a^ (23.6–34.0)	38.9 ^a^ (33.2–45.0)
No	14.0 ^a^ (11.7–16.7)	23.3 ^b^ (21.0–25.7)	17.2 ^a^ (15.4–19.2)	27.5 ^a^ (25.3–29.9)	31.2 ^a^ (28.7–33.7)	35.5 ^a^ (32.9–38.1)

IFA, iron–folic acid. Superscripts ^ab^ in the same column indicate estimates with non-overlapping CIs.

## Data Availability

The data that support the findings of this study are available from the Ministry of Health and Family Welfare (MoHFW), Government of India, but restrictions apply to the availability of these data, which were used under license for the current study, and so are not publicly available. Data are, however, available from the authors upon reasonable request and with permission of MoHFW.

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
