# Peer review of "Prevalence of Vitamin B12 and Folate Deficiencies in Indian Children and Adolescents"

_nutrients, 2023, doi:10.3390/nu15133026_

Round 1

Reviewer 1 Report

Shalini and colleagues presented findings on the prevalence of B12 and folate deficiency in children and adolescents across India, providing a comprehensive national-level perspective. The study design was appropriate, and the data analysis methods used were suitable. These results contribute to a better understanding of the B12/folate status among young individuals, which can inform healthcare policymaking. Although specific line numbers were not provided, I have some suggestions to improve certain aspects of the study.

1.       Regarding the concentrations of B12 and folate deficiency, as outlined by WHO guidelines, values below 203 pg/ml and 151 ng/ml respectively indicate deficiency. However, upon examining Figure 2, I did not observe a significant prevalence of deficiency, particularly with respect to up to 30% deficiency for both B12 and folate. It would be helpful to have the authors provide an explanation for this discrepancy.

2.       I would like to draw attention to the first sentence in section 3.2 on page 8, which compares primary schooling without considering education levels across the board. Consequently, the conclusion drawn may not be accurate. Additionally, it would be valuable to explore whether there is any connection between the quality of drinking water and folate deficiency.

3.       Furthermore, it is advisable to indent the paragraph following section 3.2. In Table 3, the authors concluded that there is no association between anthropometric undernutrition and B12 or folate deficiency. However, it is noteworthy that the groups exhibiting severe stunting in the 5-9 years age range and severe wasting in the 10-19 years age range showed significantly lower levels of B12 deficiency. It would be beneficial if the authors could provide explanations for these observations.

4.       To enhance the clarity of Figure 1, I recommend adding a first column to indicate the titles for the corresponding rows and removing any duplicated text. Moreover, Figure 2 should be bolded, and the legend should be separated from the section title "3.2." Additionally, the section labeled "3.2" on page 8 should be corrected to "3.3." In Figure 2, it may be more consistent to replace "Male/Female" with "Boys/Girls."

5.       For Table 1, aligning the first column at the top would improve clarity and separation from the subsequent columns. This alignment recommendation can be applied to other tables as well.

6.       On page 1, there is an extra "1" under the keywords section. On page 2, the term "enroll" should be replaced with "enrol." Furthermore, I would like to understand the reason behind including the statement "Ethical approval was obtained from the Institutional Review Boards of Population Council, New York, USA." Lastly, on page 3, it would be helpful to know the volume of blood drawn during the study.

7.       Include a space before and after the symbols "<" or ">".

Reviewer 2 Report

Tattari Shalini et al performed  a study of the prevalence of low blood B12 and erythrocyte folate in 1–19-year children and adolescents among states of India using the Indian Comprehensive National Nutrition Survey conducted during 2016-2018. The strength of the study is the population size and the evaluation of the influence of sex, anthropometry, demographic and socioeconomic and morbidities.

However there are a number of major and minor issues that need to be addressed. The writing style and grammar should be improved throughout manuscript.

Major  general issues

The authors should distinguish the deficits (low B12 and/or low folate) from deficiencies (with related clinical symptoms such as anemia). Their study is dealing with both categories, with a large majority of subjects with deficits. Anemia should deserve more  interest, in particular to define the two categories and evaluate its relation with blood B12 and Folate. The stratified analyses of the population according to vegetarianism is mandatory. The obesity becomes prevalent in Indian adolescents. The association between B12, Folate and low B12/high folate and insulin resistance has been reported in adult obese subjects  in Western population and in Indian children born  from mothers with low B12 and normal/high folate. It is mandatory to consider to address this issue in addition to denutrition, and the prevalence of low B12/high folate. The related data  should be commented and discussed properly in an additional paragraph of discussion section. 

Specific comments 

- Ref 1: “Stover, P. J. Physiology of folate and vitamin B12 in health and disease. Nutr. Rev. 2004, 62(6 Pt 2), S3-12; discussion S13. “

A more recent review on B12, Folate and the remethylation pathway should be cited. A review on B12 malabsorption is also needed in relation with causes of deficits in the discussion section. 

- The imbalance between B12 and folate status can predict insulin resistance in obese subjects. The author consider overweight and obesity. This should be considered in the analyses, and the results commented in the discussion according to the existing literature.  

- The paragraph “A total of 1,05,243 children and adolescents (preschool: 31,058, school-age: 38,355 and adolescents: 35,830) were interviewed and anthropometric data collected, of which serum B12 and erythrocyte FA concentrations were available for 33,880 and 38750 children and adolescents (preschool: 9,976 and 11,004, school-age: 12,156 and 14,125 and adolescents: 11,748 and 13,621, respectively) (Figure 1). The socio-demographic characteristics were almost similar among participants in whom anthropometric data were collected (total sample) and the study sample (B12 and FA), except that proportion of children included in the study sample was higher in 3-4 and 7-9y compared to 1-2 and 5-6y (61% vs 39%) re-spectively (Table S1). Table 1 shows the age-specific general characteristics of the study population. Among preschool children, 35% were stunted and underweight, 16% were wasted, and about 15% had diarrhea two weeks prior to the survey in both B12 and FA study sample.” should be moved as first paragraph in result section 

- Data should be also analyzed according to low B12/normal vs high folate and commented in a specific paragraph in discussion section 

- Results should be reported in pmol/L for B12 and nmol/g for folate 

- Tables 2A and 2B report the increase of either B12 and folate deficit among age. 

The subjects with combined deficit B12 and folate and those with low B12/ high folate should be also reported. Since the distribution of serum B12 and erythrocyte folate is not symmetrical, I suggest to report also the median and inter quartiles 

- “adolescent boys had 8% points and 5% points higher B12 and FA deficiency compared to girls” Not clear, please rephrase 

- Paragraph “The point prevalence of B12 deficiency varied across the states:” a geographic map showing the states with higher, medium and lower  prevalence of either B12 or Folate (with use of specific colours) should be helpful. 

- “The point prevalence of B12 deficiency varied across the states:…(Figure 4).” Figure 4 is misquoted 

- “Table 4. Prevalence of vitamin B12 and folate deficiency with IFA supplementation in all age groups” The authors should avoid abbreviations (IFA) in the title 

- Discussion, first sentence. The authors should avoid any statement such as “This is the first study from India providing the serum B12 and erythrocyte FA levels and their prevalence estimates, in a representative sample of children and adolescents, at the national, state and regional level.” but rather point out the key contributions of their study to the field in the beginning of discussion. 

- “Nationally representative surveys in other countries have also demonstrated a low prevalence of B12 deficiency among 1-6 y children in Mexico (7.7%) [29], and school-aged children in Venezuela (11.4%) [30,31], although a Guatemalan study reported the prevalence deficiency of B12 and FA to be 22.5 and 33.5% respectively, among children aged 6-59 months [6].” Unclear, please rephrase. The stratification between deficiency and depletion does not correspond to established categorizations of B12 status 

- “The differences in the magnitude of the prevalence of deficiency among the earlier studies might be multifactorial including methodological issues.” Rephrase, unclear 

- “For B12, it has been argued that its measurement lacks sensitivity or specificity and that biomarkers …..holo-transcobalamin compared to total B12 measurement (37%) [11]. These differences make it difficult to choose an appropriate threshold to define deficiency.” The whole paragraph should be reworded. The aim of the study is not the accuracy of biologicals parameters to assess B12 deficiency but the prevalence of B12 and folate deficits. The paragraph should be rather focused in limitations as the authors could not assess  MMA, Hcy and/or holoTC to correctly evaluate the B12 status and stratify cases according to deficits (with abnormal blood markers) and deficiencies (with clinical symptoms)

- “Although deficiency of these vitamins can occur primarily as a result of insufficient dietary intake or malabsorption,” and “Rural or urban residence represents an aggregate of multiple factors” Vegetarianism is a major issue that is not properly considered in this study in regard to its high prevalence and relation to religion. These two determinants should be included in the study design by performing stratified analyses and looking for their influence in the geographical and socioeconomical determinants. The data should be discussed and  literature properly cited (at least a recent umbrella review of systematic analyses and meta-analyses  published in Clinical Nutrition in 2020) 

- “Adolescent boys had higher prevalence of B12 deficiency” and “may be explained by a higher requirement of micronutrients among them to sustain rapid muscular growth during adolescence, as compared to girls”

The decreased of food diversity intake is also reported in adolescence, including in those from  higher social categories. This should be considered. It could also explain the paradoxical higher prevalence of FA deficiency in participants from higher socio-economic status. 

- “Another limitation pertains to the lack of data on other biomarkers such as MMA, Hcy and holotranscobalamin.” This limitation needs to be commented more extensively 

The writing style and grammar should be improved throughout manuscript.

Round 2

Reviewer 1 Report

Thanks for the responses.

For comment 1, I’m asking about the discrepancy between the concentration of B12/folate and the WHO cutoff (203 pg/ml and 151 ng/ml, respectively). The concentrations reported here are much higher than the guideline. How do authors define the deficiency?

Author Response

Response: The authors are so thankful to the reviewer for the prudent evaluation of the manuscript with the very constructive suggestions that helped to improve the manuscript.

Regarding the comment about the discrepancy between the concentration of B12/folate and the WHO cutoffs, our sincere apologies for the way that we understood and answered it (in the 1st round) instead of what the reviewer expected us to clarify the discrepancy. As rightly highlighted by the reviewer, the reported concentrations are higher than the deficiency cutoff values (203 pg/ml and 151 ng/ml, respectively for B12 and FA). The reported concentrations are geometric means of the given age band (Table 2) or the given age (Figure 2) for the given sample number of that age band or age. The learned reviewer surely knows that mean concentration includes both normal and deficient values and the mean value (concentration) will not indicate the extent of deficiency. We get the prevalence of deficiency only when we calculate the proportion of the samples (out of the total samples) that have a concentration below the cutoff (203 pg/ml and 151 ng/ml). The discrepancy between the concentration and the cutoff values of B12/folate that the reviewer is referring to might be this and we believe this explanation clears the concern of the reviewer.